# Quality, Equity and Utility of Observational Studies during 10 Years of Implementing the Structured Operational Research and Training Initiative in 72 Countries

**DOI:** 10.3390/tropicalmed5040167

**Published:** 2020-11-06

**Authors:** Rony Zachariah, Stefanie Rust, Pruthu Thekkur, Mohammed Khogali, Ajay MV Kumar, Karapet Davtyan, Ermias Diro, Srinath Satyanarayana, Olga Denisiuk, Johan van Griensven, Veerle Hermans, Selma Dar Berger, Saw Saw, Anthony Reid, Abraham Aseffa, Anthony D Harries, John C Reeder

**Affiliations:** 1United Nations Childrens Fund (UNICEF), United Nations Development Programme (UNDP), World Bank, World Health Organization (WHO), Special Programme for Research and Training in Tropical Diseases (TDR), 1202 Geneva, Switzerland; khogalim@who.int (M.K.); armidiea@who.int (A.A.); reederj@who.int (J.C.R.); 2Koninklijke Nederlandse Centrale Vereniging tot bestrijding der Tuberculose (KNCV) TB Foundation, 2516AB The Hague, The Netherlands; steffi.rust87@gmail.com; 3Centre for Operational Research, International Union Against Tuberculosis and Lung Disease, 75006 Paris, France; pruthu.tk@theunion.org (P.T.); akumar@theunion.org (A.M.K.); ssrinath@theunion.org (S.S.); sberger@theunion.org (S.D.B.); adharries@theunion.org (A.D.H.); 4The Union South East Asia Office, New Delhi 110016, India; 5Yenepoya Medical College, Yenepoya (Deemed to Be University), Mangaluru 575018, India; 6Tuberculosis Research and Prevention Center NGO (TB-RPC), Yerevan 0014, Armenia; davkaro@gmail.com; 7Department of Medicine, University of Gondar, Gondar 6200, Ethiopia; Ermi_diro@yahoo.com; 8Alliance for Public Health, 01601 Kyiv, Ukraine; o.denisiuk@yahoo.com; 9Institute of Tropical Medicine, 2000 Antwerp, Belgium; jvangriensven@itg.be; 10Médecins Sans Frontières, Qperational Centre Brussels, 1617 Luxor, Luxembourg; Veerle.HERMANS@luxembourg.msf.org (V.H.); tony.reid@brussels.msf.org (A.R.); 11Department of Medical Research, Pyin Oo Lwin 05085, Myanmar; sawsawsu@gmail.com; 12London School of Hygiene and Tropical Medicine, London WC1 7HT, UK

**Keywords:** STROBE, operational research, SORT IT, observational studies, universal health coverage, health systems research

## Abstract

**Introduction:** Observational studies are often inadequately reported, making it difficult to assess their validity and generalizability and judge whether they can be included in systematic reviews. We assessed the publication characteristics and quality of reporting of observational studies generated by the Structured Operational Research and Training Initiative (SORT IT). **Methods:** A cross-sectional analysis of original publications from SORT IT courses. SORT IT is a global partnership-based initiative aimed at building sustainable capacity for conducting operational research according to country priorities and using the generated evidence for informed decision-making to improve public health. Reporting quality was independently assessed using an adapted version of ‘Strengthening the Reporting of Observational Studies in Epidemiology’ (STROBE) checklist. **Results:** In 392 publications, involving 72 countries, 50 journals, 28 publishers and 24 disease domains, low- and middle-income countries (LMICs) first authorship was seen in 370 (94%) and LMIC last authorship in 214 (55%). Publications involved LMIC-LMIC collaboration in 90% and high-income-country-LMIC collaboration in 87%. The majority (89%) of publications were in immediate open access journals. A total of 346 (88.3%) publications achieved a STROBE reporting quality score of >85% (excellent), 41 (10.4%) achieved a score of 76–85% (good) and 5 (1.3%) a score of 65–75% (fair). **Conclusion:** The majority of publications from SORT IT adhere to STROBE guidelines, while also ensuring LMIC equity and collaborative partnerships. SORT IT is, thus, playing an important role in ensuring high-quality reporting of evidence for informed decision-making in public health.

## 1. Introduction

Operational research is vital to improve the quality and performance of health care delivery, especially in low- and middle-income countries (LMICs) [1]. Aimed at building the science of solutions, operational research is conducted close to the supply and demand of health services and has a key role in achieving universal health coverage (UHC) [2].

Operational research is often observational in nature and including cross-sectional, cohort and case-control designs. Increasingly, mixed methods and qualitative research designs are being used. Such studies help address real-world questions that cannot be answered by randomized controlled trials [3,4], such as: what are treatment outcomes under operational conditions; are there late side-effects of medications; do behavioral aspects affect uptake of health interventions [5]? Operational research is also essential in finding out “how to” deliver new tests, vaccines and drugs that save lives during outbreaks [6]. The world health organization (WHO) increasingly relies on such evidence for informing its implementation guidelines [7].

Despite the importance of observational studies, they are often inadequately or incompletely reported, making it difficult to assess their validity and generalizability and to judge whether they can be included in evidence synthesis and systematic reviews that guide policy and practice change [8,9,10,11].

The Structured Operational Research and Training Initiative (SORT IT) is a proven model for strengthening the operational research capacity in LMICs and has been described in previous publications [8,12,13]. In brief, SORT IT is a global partnership-based initiative that is coordinated by TDR, the Special Program for Research and Training in Tropical Diseases [14] and has over 50 implementing partners, including disease control programs, non-governmental organizations and academia. SORT IT builds sustainable capacity to conduct operational research according to country priorities and uses the generated evidence for informed decision-making to improve public health. Put simply, it aims to make countries “data rich, information rich and action rich.” Between 2009 and 2019, SORT IT was scaled up to 93 countries and published over 550 scientific papers with about 70% self-reporting an influence on policy and/or practice [8,15,16].

Mindful of the importance of ensuring high reporting standards, SORT IT included the STROBE (Strengthening the Reporting of Observational Studies in Epidemiology) guidelines [5] as an “integral part of its training.” The STROBE checklist allows authors and independent observers to verify whether various items have been reported in a paper and to assess how complete that reporting is against a “gold standard.” The checklist can then be used to calculate a score for grading quality of reporting [5]. The importance of ensuring reporting quality has been underscored by authors in the British Medical Journal MJ [17] and the bulletin of the World Health Organization [7].

In addition to assessing reporting quality, we believe that operational research in LMICs should be assessed through a wider lens of (a) how relevant the research is to the local context, (b) whether there is LMIC leadership and gender-equity in authorship, (c) whether gender-disaggregated analysis is included to identify gender-based inequalities in access to health services, (d) whether there are high-income-countries (HIC)-LMIC and LMIC-LMIC collaborations, which are highly important for building communities of practice and e) whether the research is published in a timely manner in an open access journal to facilitate global access to generated knowledge [18].

Assessing SORT IT publications through this wider perspective provides a yardstick for assessing equity, utility and quality. A PubMed search revealed a few studies evaluating the quality of reporting of observational studies, but these were restricted to specific themes such as otorhinolaryngology [19], dermatology [20], occupational medicine [21] and Ebola [22]. No studies have so far assessed the quality of reporting in publications stemming from a global research capacity building program and covering different public health domains. We, thus, aimed to assess the quality of publications generated over a decade through the global SORT IT program.

Specific objectives were to determine (a) the publication characteristics, including LMIC leadership in authorship and collaborative partnerships; (b) timeliness, type of journal access and usage metrics (citations and downloads); and (c) quality of reporting using an adapted STROBE checklist.

## 2. Materials and Methods

### 2.1. Study Design

This was a cross-sectional study involving an analysis of original research articles published in peer-reviewed journals from SORT IT courses in Africa, Asia, Latin America, Europe, Central Asia and the South Pacific.

### 2.2. The SORT IT Program

Some of the characteristics of SORT IT courses, such as participant selection criteria, milestones and outputs, have been described previously [12,13]. In brief, the training has three modules scheduled over 10 to 12 months: module 1 (5–6 days) on research protocol development, module 2 (5–6 days) on efficient data capture and data analysis and module 3 (5–7 days) on how to write a scientific paper. To move from one module to the next, participants must achieve certain milestones. A participant is judged to have successfully completed SORT IT if they reach all milestones, including the final submission of a completed manuscript to a peer-reviewed journal, within four weeks of completing module 3. The SORT IT model is unique in that it simultaneously combines research training with research implementation (an apprenticeship approach) that empowers participants. There is also an inbuilt system for training the trainers. SORT IT also formally follows up participants for up to 18 months after course completion to assess whether the acquired skills have been further utilized and if the research has contributed to changes in policy and practice.

### 2.3. Publication Characteristics Including Authorship and Collaborative Partnerships

The publication themes, authorship details including first/last authors from LMIC and HIC, and the corresponding author were derived from the SORT IT database. The number and types of institutional affiliations (HIC, LMIC) were sourced from the title page of each published paper. For the purposes of this study, an HIC-LMIC partnership means the principal investigator was affiliated to an institution in an HIC, but authors from an LMIC were included as co-authors. In an LMIC-LMIC partnership, at least one co-author beside the principal investigator was from another LMIC country and institution. Understandably, in the latter, authors from an HIC may also be involved.

### 2.4. Time to Publication, Access Type and Article Usage Metrics

“Time to publication” was calculated by determining the interval between date of first submission and the date of eventual publication (even if this was to a different journal if the paper was rejected by the first journal). Date of submission was indicated in the SORT IT database and date of publication on the final published paper. Information on access (immediate open access, delayed open access—often following an embargo period of 6–12 months—or subscription-based access) was obtained from the journal website [18]. Article views, citations and downloads were sourced from the journal site. If these were unavailable, it was noted.

### 2.5. Completeness of Reporting in Line with the STROBE Statement

The standard STROBE checklist includes 22 items that relate to the title, abstract, introduction, methods, results and discussion sections of published papers [5]. We included two additional items: (a) local relevance of the research question (i.e., will the study have operational implications or have the potential of leading to policy and practice?) as indicated in the publication and (b) presence of an ethics statement which we felt is indispensable to good-quality operational research [23]. Our modified checklist with 24 items (including instructions for use) is shown in the Appendix A.

The modified checklist was pilot-tested on a sample of 20 papers by two independent reviewers who were experienced in critical appraisal techniques and familiar with STROBE. Each reviewer scored each article independently. In order to maximize the reliability between reviewers, reviewer scores were compared for each paper and cross-validated and any outstanding disagreement over items was resolved by adjudication with a third senior reviewer. This helped in arriving at a common understanding of the checklist and the assessment methods. The same process was then used to grade all the papers.

For each paper, a reporting score was derived by dividing the number of adequately reported items (the numerator) by the 24 items in the modified checklist (the denominator) and this was expressed as a percentage. Every item received a score of 1 (if reported) or 0 (if not reported) or 0.5 if only one of two sub-items were reported. Similarly, scores of 0.2, 0.25 or 0.33 were given depending on how many sub-items were applicable. For items 1a and 1b, a positive response was scored as 0.5 each. Items 12, 13, 14 and 16 had multiple components, each of which might or might not have been applicable. The total applicable components were considered in scoring. For example, item 12 had components 12a to 12e. If both components 12a and 12b were applicable to a given study and both were reported, the score was 1. If only 12a was reported and not 12b, this was considered as a score of 0.5. As all 24 items were not applicable to each paper, we recalculated the percentage using the total number of “applicable items” as denominator. To limit reviewer bias, the two reviewers were independent and not part of the TDR repository of SORT IT facilitators/mentors.

For each paper, reporting quality was graded as: <65% unsatisfactory; 65–75% = fair; 76–85% good; >85% = excellent [24]. The inter-reviewer agreement (cohen’s kappa) was calculated to determine if there were large differences in the scoring of items between the two reviewers. A kappa score over 0.75 represents excellent agreement beyond chance, a kappa of 0.40–0.75 represents intermediate to good agreement and a kappa below 0.40 represents poor agreement. The scoring data was entered in Microsoft Excel.

### 2.6. Study Inclusion and Period

This study included all observational studies (original articles) involving quantitative data that stemmed from SORT IT courses that were initiated and completed between 1st January 2009 and 31st December 2018. The censor date was deliberately set as subsequent to this date, TDR has actively embarked on franchising SORT IT courses to various implementing partners around the globe. Information on the quality of publications up to and including this date thus serves as “a baseline,” against which subsequent sets of publications can be assessed and compared in a similar manner. The setting of this censor date also aligns with the TDR scientific advisory committee recommendation to have a yardstick, against which future audits of publications can be compared as the SORT IT brand is franchised and rolled out by implementing partners in efforts towards achieving universal health coverage.

Viewpoints, short reports and mixed/qualitative research studies were excluded. The study was conducted over a one-year period from April 2019 to March 2020.

### 2.7. Data Sources, Collection and Statistical Analysis

The SORT IT monitoring database (Microsoft Excel), which contains information on all SORT IT publications, was used to extract the data of observational studies. A new database was created and additional variables were added, including: (a) institution of the principal investigator, (b) number of institutions from HICs, (c) number of institutions from LMICs, (d) disease category, (e) presence of sex-disaggregated analysis, (f) time to publication, (g) access type, h) journal impact factor, (i) number of article views, (j) number of article citations, (k) number of article downloads and (l) STROBE reporting score. Results were reported using numbers and proportions; 95% confidence intervals were used where appropriate and a *p*-value < 0.05 was considered statistically significant.

### 2.8. Ethics

Ethics approval was obtained from the ethics advisory group of the International Union against Tuberculosis and Lung Disease, Paris, France (with whom TDR has a formal agreement for ethics reviews on observational studies) and the ethics review board of Médecins Sans Frontières, Geneva, Switzerland. As the study included publications already in the public domain, the issue of individual informed consent did not apply.

## 3. Results

### 3.1. SORT IT Courses and Publication Outputs

Of 50 completed SORT IT courses, three involved qualitative designs and were excluded. Country course locations of the remaining 47 courses included: Africa—Ethiopia (4), Kenya (4), Zimbabwe (2), Liberia (1), Sierra-Leone (1) and Uganda (1); Asia—India (7), Nepal (4), Sri-Lanka (2), Pakistan (2), Myanmar (3) and Kazakhstan (1); Europe—France (5), Luxembourg (4) and Estonia (1); South America—Panama (1); South Pacific—Fiji (4).

A total of 529 manuscripts were submitted to peer-reviewed journals, of which 403 (76%) were published by 31 December 2018 (the censor date). Eleven publications involved qualitative, mixed-methods designs and opinion pieces and were excluded, leaving a total of 392 included in the analyses.

### 3.2. Publications, Authorship Characteristics and Collaborative Partnerships

Of 392 published studies, 207 (53%) were cross-sectional, 179 (46%) were cohort and 6 (1.5%) were case-control designs. Sex-disaggregated analysis was included in 204 (52%) publications.

Figure 1 shows the geographic distribution of published research projects in 72 countries. Francophone Africa was relatively sparse in publications. Publications included 24 disease themes, the 10 most frequent being tuberculosis (45%), HIV/AIDS (13%), malaria (7%), non-communicable diseases (7%), maternal and child health (7%), Ebola/other outbreaks (5%), neglected tropical diseases (3%), trauma and health emergencies (3%), access to care (2%), cancers (2%) and 1% each for mental health, community involvement, nutrition, tobacco, water and sanitation and antimicrobial resistance.

Table 1 shows the authorship characteristics and collaborative partnerships involved in the 392 publications. There was LMIC first authorship in 370 publications (94%) and last authorship in 214 (55%) publications. Front-line health workers (implementers) from disease control programs, ministries of health and non-governmental institutions comprised the majority (79%) of first authors. Of 31 publications where the principal investigator was from an HIC, 27 (87%) involved HIC-LMIC collaboration. Similarly, of 361 publications where the principal investigator was from an LMIC, 325 (90%) involved LMIC-LMIC collaboration. An average of 2.2 (IQR: 1–3) institutions from an HIC and 3.6 (IQR 2–5) institutions from an LMIC were represented on each publication.

### 3.3. Time to Publication, Article Access Type and Usage Metrics

Papers were published in 50 journals (journal impact factor ranging from 0–19) and involved 28 publishers. The median time to publication was 6.2 months from first submission (inter-quartile range = 5.1–9.9 months; range 0.5–48 months).

The majority (89%) of publications were in 37 journals that provided immediate open access and the remaining (11%) were in 13 delayed open access or subscription-based journals (Table 1).

Availability of altimetric measures on the journal site varied between journals. Of all 392 papers, only 39% had data on article views, 39% had data on citation metrics and 24% had data on article download metrics (Table 1). Only 46% of open access journals and 30% of other journals presented one or more of the three altimetric measures (views, citations, downloads). Article views were 3.7-fold higher in open access journals compared to other journals. Similarly, mean citations were higher for open access (6.6 per open access paper) compared to other papers (4.0 per paper).

### 3.4. Completeness of Reporting in the STROBE Checklist

The inter-rater reliability between reviewers was 0.83 (kappa statistic). There was only a 1% overall difference in scores between reviewers, and after reconciliation with the senior reviewer the STROBE scores from the reviewer with the lower overall score are presented.

Table 2 shows the quality of reporting of the 392 published papers in relation to the adapted STROBE checklist. A total of 346 (88.3%) individual papers achieved a STROBE score of >85% (excellent), 41 (10.4%) achieved 76–85% (good) and 5 (1.3%) achieved a score of 65–75% (fair). None were categorized as unsatisfactory (<65%).

Table 3 shows the percentage by sub-items in the STROBE checklist which were reported, not reported and not applicable. The mean of reported sub-items was 74%. Of the 49 sub-items, 18 sub-items (namely 1a, 1b, 2–4, 5a, 7–11 and 18–24) should have been reported on all papers irrespective of the study design. Sixteen of these items were reported in >90% of papers. Two sub-items were indicated in less than 90% of papers and included: indicating the study design in the title or abstract (1a) and efforts made to reduce potential bias (9). Ten of the 18 items mentioned above were reported in all publications.

## 4. Discussion

This study involves the largest dataset of observational studies ever assessed for quality of reporting, and included 72 countries, 50 journals and 24 publication domains. Almost nine in every ten publications were graded as being of excellent reporting quality according to STROBE guidelines. LMIC first authorship was present in 90% of publications, a female first author was present in about half of all publications and about 90% of papers included HIC-LMIC and LMIC-LMIC partnerships. These findings showcase the important role SORT IT is playing in generating high-quality evidence while ensuring equity and collaborative partnerships.

This assessment provides reassurance that the majority of publications generated by the SORT IT global partnership is reported to a high-quality level and can be used for informed decision-making in public health.

The study is also important as it can impact practice by providing a baseline for future comparisons of publications done by implementing partners who take up the SORT IT brand; it can impact the quality of research per se by enhancing adherence to STROBE guidelines and, importantly, it provides reassurance for observational studies to be included in evidence synthesis for the development of guidelines that may impact policy and practice.

The additional finding that 79% of publications were led by front-line health workers indicates that the evidence was generated from those who work close to the supply and demand of health services. Such evidence from the front-lines of public health is important in efforts towards achieving universal health coverage [2]. In general, observational studies are seldom used for policy decisions because they are often considered “lower-level evidence,” as they are often inadequately reported. A worthwhile point is that these studies are often based in local, front-line settings, and if properly done and reported, should be considered as part of the evidence base for policy decisions. This would close the loop on the uptake of evidence from these studies.

The study strengths are that all published observational studies from a decade of work were included, limiting the risk of selection bias; each paper was independently assessed by two experienced reviewers; and data on gender, LMIC equity and collaborative partnerships were sourced from a robust data base used for quarterly reporting of SORT IT performance targets. Although not included in the STROBE checklist, we also assessed if ethics approval was included in each publication and we advocate for its inclusion in the STROBE checklist [23]. We also assessed if the local relevance of research—an indicator of “homegrown” research—was included. John Walley et al. eloquently summarized the relevance of the latter: “if you want to get research into practice, first get practice into research” [4].

There were some limitations. The fixed censor date of 31 December 2018 obviously excluded papers published in 2019 and 2020. However, as stated in the methods section, this was a deliberate decision to provide a yardstick for SORT IT publications between 2009 and 2019, against which future publications could be compared as the SORT IT brand is franchised and rolled out by other implementing partners. Our assessment using altimetric data is also limited, as we relied on what was available on the journal websites, and did not include other portals such as Google Scholar, ResearchGate or institutional repositories. This was deliberate, as we aimed to identify journals which provided readily accessible altimetric data. Finally, in addition to assessing each paper for reporting quality and providing an overall STROBE score, we also provided a summary measure of reported sub-items by calculating a mean percentage. Since the values of all sub-items are not really the same, this is not ideal, but it allows readers to make an overall comparison with other studies that have used a similar methodology.

These limitations notwithstanding, the study findings have some important implications. First, the proportion of publications graded as having an excellent cumulative STROBE score in our study (88%) and the mean score for reporting all sub-items (74%) exceeds that in the literature. A study by Hendriksma et al., which included the top five general medical journals (New England Journal of Medicine, The Lancet, Journal of the American Medical Association, British Medical Journal and PLoS Medicine), revealed a mean STROBE score of 69%, while that in five otorhinolaryngologic journals achieved a mean score of 51% [19]. Another study, including observational studies published in prestigious high impact journals of occupational medicine, two of which had endorsed STROBE in their author guidelines [21], described 64% reporting of all sub-items. In a study by Poorolajal et al. [25], which included cohort studies published in the six top scientific medical journals (New England Journal of Medicine, Journal of the American Medical Association, The Lancet, British Medical Journal, Archives of Internal Medicine and Canadian Medical Association Journal), the total percentage of sub-items reported was 69%. Since Poorolajal et al. focused on cohort studies, which are more expensive and needing longer follow up, a higher STROBE score might have been expected.

The question, therefore, is: why is the quality of reporting in SORT IT substantially higher? The reasons are intuitive and include: endorsement and intrinsic embedding of the STROBE checklist as part of the training on protocol and manuscript writing; the fastidious apprenticeship approach to training that involves hands-on mentorship and critical appraisals [26]; and the inclusion of the STROBE checklist in publications as a routine indicator of SORT IT performance standards.

While it is understandable that certain sub-items (e.g., sensitivity analysis) may not be applicable to some studies (as they are not part of their study objectives), we still identified areas where further improvements could be made, such as including the study design in the title or abstract and highlighting efforts to reduce bias. Some of this may be difficult to implement in practice, as journal author guidelines may restrict the number of characters in the title of a paper. Nevertheless, we call on SORT IT partners to take on these ideas to further improve reporting of SORT IT-related research.

Second, the median time to publication was 6.2 months, implying that SORT IT evidence is timely. This might be explained by the excellent overall quality of SORT IT papers and the continued hands-on mentorship during the peer review-response process that reduces time spent in repeated peer reviews prior to publication.

Third, and most noteworthy, is the prominent (92%) first authorship position from LMICs (including female authors), which is in stark contrast to the usual under-representation seen in leading medical journals. For example, in a study of authorship on 236 articles in the Lancet Global Health on research in LMICs, only 35% of the authors were affiliated with, or came from LMICs [27]; LMIC authorship should be considered a proxy for research leadership and deserves attention as an indicator of the success of capacity-building programs [28]. It is also a good indicator of respectful team dynamics which the SORT IT partnership is currently achieving [29].

Fourth, the low availability of article altimetric data on the journal websites (views, citations and downloads) in both open access journals (46%) and subscription-based/delayed access journals (30%) is far from desirable. We strongly believe that all journals should provide authors with access to altimetric data in a standardized manner to allow judgements on research utility without the need to spend time searching other sources (Google Scholar, ResearchGate or institutional repositories). Commendable examples which provide such data include the PloS, BMC and MDPI journals.

Finally, we take note of the relative sparseness of publications from francophone Africa, which would be related to English language barriers in a publishing world that is anglophone-dominated. There is an imperative for pro-active steps to address this gap. Attempts to improve gender disaggregated analysis should also be taken on board.

In conclusion, this study highlights the important role SORT IT has played in generating high quality reporting of evidence for informed decision-making in public health while ensuring LMIC equity, engagement and collaborative partnerships. This study will be a yardstick for future audits and ensuring vigilance, while SORT IT is being franchised in efforts towards achieving universal health coverage.

## Figures and Tables

**Figure 1 tropicalmed-05-00167-f001:**
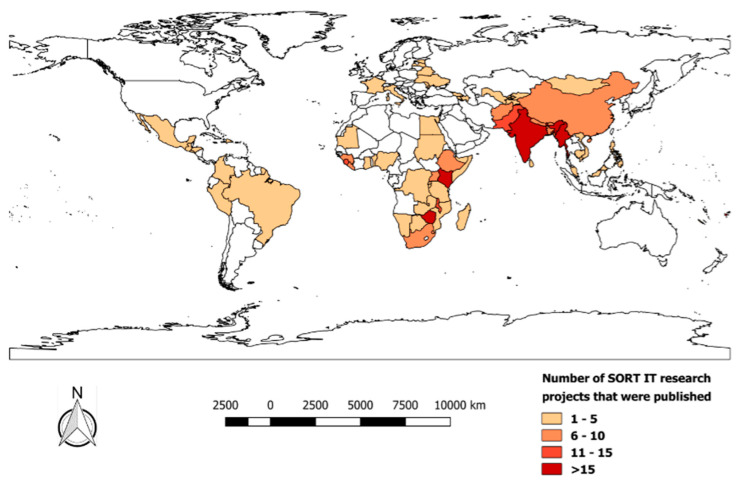
Global distribution of 392 publications from 47 completed Structured Operational Research and Training Initiative courses (January 2009–December 2018).

**Table 1 tropicalmed-05-00167-t001:** Characteristics of published papers from 47 completed Structured Operational Research and Training Initiative courses (January 2009–December 2018).

Characteristics of Published Papers	*N*	(%)
**Total Publications**	392	
**Number of Journals**	50	
**Journal Impact Factor** (Range)	0–19	
**LMIC Author**		
First author	370	(94)
Corresponding author	367	(94)
Last author	214	(55)
**Gender Equity**		
Female first author	173	(44)
**Affiliation of the First Author**		
Disease control programs/Ministries of Health	180	(46)
International or national NGOs	133	(33)
Academic institutions	63	(16)
WHO	12	(3)
Donors	4	(1)
**Institutions**		
Mean number from HICs (range)	2.2	(0–8)
Mean number from LMICs (range)	3.6	(0–9)
HIC-LMIC partnerships (*n*−31) ^1^	27	(87) ^1^
LMIC-LMIC partnerships (*n*−361) ^2^	325	(90) ^2^
**Sex-Disaggregated Analysis**		
Included in background and/or analysis	204	(52)
**Journal Access Type**		
Immediate open access	349	(89)
Delayed open access	13	(3)
Subscription-based	30	(8)
**Article Views**		
Papers with article view metrics	153	(39)
Total views	271428	
**Article Downloads**		
Papers with article download metrics	95	(24)
Total downloads	9127	
**Article Citations**		
Papers with article citation metrics	153	(39)
Total citations	998	

^1^ High income country (HIC)—low- and middle-income country (LMIC) partnership was when the principal investigator was from an institution in an HIC (*n*−31) but authors from an LMIC were included as co-authors. ^2^ An LMIC-LMIC partnership was when the principal investigator was from an LMIC (*n*−361) and there was also a co-author(s) from at least one different institution from another LMIC country other than that of the principal investigator.

**Table 2 tropicalmed-05-00167-t002:** Completeness of reporting of publications from 47 completed Structured Operational Research and Training Initiative courses according to an adapted STROBE checklist (January 2009–December 2018).

Completeness of Reporting	*N*	(%)
**Total Publications**	392	
**STROBE Scores ^a^**		
>85% (Excellent)	346	(88.3)
76–85% (Good)	41	(10.4)
65–75% (Fair)	5	(1.3)
<65% (Unsatisfactory)	0	(0)

STROBE: Structured Reporting of Observational Studies in Epidemiology. ^a^ Assessed on 24 items in an adapted STROBE checklist.

**Table 3 tropicalmed-05-00167-t003:** Percentage of items in the STROBE checklist which were reported in 392 studies (179 cohort, six case control and 207 cross-sectional studies) from the Structured Operational Research and Training Initiative (2009–2018).

Item	Recommendation	Reported n (%)	Not Reported n (%)	Not Applicable n (%)
1a	(a) Indicate the study’s design with a commonly used term in the title or the abstract	264 (67.3)	128 (32.7)	0
1b	(b) Provide in the abstract an informative and balanced summary of what was done and what was found	390 (99.5)	2 (0.5)	0
2	Explain the scientific background and rationale for the investigation being reported	392 (100)	0	0
3	State specific objectives, including any pre-specified hypotheses	391 (99.7)	1 (0.3)	0
4	Present key elements of study design early in the paper	392 (100)	0	0
5a–g	Describe: setting, locations, relevant dates, periods of recruitment, exposure, follow-up, data	392 (100)	0	0
6a	(a) Cohort study—Give the eligibility criteria, and the sources and methods of selection of participants. If applicable, describe methods of follow-up	179 (100)	0	0
6b	Case-control study—Give the eligibility criteria, and the sources and methods of case ascertainment and control selection. Give the rationale for the choice of cases and controls	6 (100)	0	0
6c	Cross-sectional study—Give the eligibility criteria, and the sources and methods of selection of participants if applicable	206 (99.5)	1 (0.5)	0
6d	(b) Cohort study—For matched studies, give matching criteria and number of exposed and unexposed	0	0	179 (100) ^1^
7	Clearly define all outcomes, exposures, predictors, potential confounders, and effect modifiers. Give diagnostic criteria, if applicable	388 (99.0)	4 (1.0)	0
8	For each variable of interest, give sources of data and details of methods of assessment (measurement). Describe comparability of assessment methods if there is more than one group	390 (99.5)	2 (0.5)	0
9	Describe any efforts to address potential sources of bias	345 (88.0)	47 (12.0)	0
10	Explain how the study size was determined	391 (99.7)	1 (0.3)	0
11	Explain how quantitative variables were handled in the analyses. If applicable, describe which groupings were chosen and why	387 (98.7)	0	5 (1.3)
12a	(a) Describe all statistical methods, and if applicable, those used to control for confounding	304 (77.6)	4 (1.0)	84 (21.4)
12b	(b) If applicable, describe any methods used to examine subgroups and interactions	204 (52.0)	13 (3.3)	175 (44.7)
12c	(c) If applicable, explain how missing data were addressed.	136 (34.7)	69 (17.6)	187 (47.7)
12d	(d) Cohort study—If applicable, explain how loss to follow-up was addressed	51 (28.5)	59 (33.0)	69 (38.5)
12e	Case-control study—If applicable, explain how matching of cases and controls was addressed	4 (66.7)	2 (33.3)	0
12f	Cross-sectional study—If applicable, describe analytical methods taking account of sampling strategy	29 (14.0)	6 (2.9)	172 (83.1)
12g	(e) If applicable, describe any sensitivity analyses	10 (2.6)	1 (0.3)	381 (97.1)
13a	(a) Report numbers of individuals at each stage of study—e.g., numbers potentially eligible, examined for eligibility, confirmed eligible, included in the study, completing follow-up, and analyzed	354 (90.3)	0	38 (9.7)
13b	(b) If applicable, give reasons for non-participation at each stage	42 (10.7)	6 (1.5)	344 (87.8)
13c	(c) If applicable, consider use of a flow diagram	105 (26.8)	8 (2.0)	279 (71.2)
14a	(a) Give characteristics of study participants (e.g., demographic, clinical, social) and information on exposures and potential confounders	346 (88.3)	1 (0.3)	45 (11.4)
14b	(b) If applicable, Indicate number of participants with missing data for each variable of interest	225 (57.4)	25 (6.5)	142 (36.1)
14c	(c) Cohort study—If applicable, summarize follow-up time (e.g., average and total amount)	92 (51.4)	20 (11.2)	67 (37.4)
15a	Cohort study—Report numbers of outcome events or summary measures over time	179 (100) ^1^	0	0
15b	Case-control study—Report numbers in each exposure category, or summary measures of exposure	6 (100) ^1^	0	0
15c	Cross-sectional study—Report numbers of outcome events or summary measures	196 (95) ^1^	0	11(5)
16a	(a) If applicable, give unadjusted estimates and, if applicable, confounder-adjusted estimates and their precision (e.g., 95% confidence interval). Make clear which confounders were adjusted for and why they were included	286 (73.0)	2 (0.5)	104 (26.5)
16b	(b) If applicable, report category boundaries when continuous variables were categorized	176 (45.0)	1 (0.3)	215 (54.7)
16c	(c) If applicable and relevant, consider translating estimates of relative risk into absolute risk for a meaningful time period	25 (6.4)	5 (1.2)	362 (92.4)
17	Report other analyses done—e.g., analyses of subgroups and interactions, and sensitivity analyses	75 (19.1)	7 (1.8)	310 (79.1)
18	Summarize key results with reference to study objectives	392 (100)	0	0
19	Discuss limitations of the study, taking into account sources of potential bias or imprecision. Discuss both direction and magnitude of any potential bias	388 (99.0)	4 (1.0)	0
20	Give a cautious overall interpretation of results considering objectives, limitations, multiplicity of analyses, results from similar studies, and other relevant evidence	392 (100)	0	0
21	Discuss the generalizability (external validity) of the study results	355 (90.5)	37 (9.5)	0
22	Give the source of funding and the role of the funders for the present study and, if applicable, for the original study on which the present article is based	390 (99.5)	2 (0.5)	0
23	Local relevance of the research question indicated/mentioned in the paper	392 (100)	0	0
24	Ethics statement included	391 (99.8)	1 (0.3)	0
	Total Percentage	**74.0%**	**3.4%**	**23%**

^1^ Applicable only to cohort studies (*n* = 179) or cross-sectional studies (*n* = 207) or case control studies (*n* = 6).

## Data Availability

De-identified study data are available on reasonable request from the corresponding author (zachariahr@who.int). A justification for its further use should be provided.

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
