# Peer review of "Quality, Equity and Utility of Observational Studies during 10 Years of Implementing the Structured Operational Research and Training Initiative in 72 Countries"

_tropicalmed, 2020, doi:10.3390/tropicalmed5040167_

Round 1
Reviewer 1 Report
Thank you very much for your submission and the opportunity to review.
My major concerns are as follows:
- Kindly ensure all references are formatted properly- please check line by line and word by word (e.g. spell WHO in reference author list)- many issues here.
- 2. The study period should be extended to 2020. Currently, you only have reviews till 2018. The data needs to be up to date and more relevant so include up to 2020.
- The data analysis seems to be mostly descriptive- please try ways for more sophisticated data analysis by demographic or regional differences.
- It would also be pertinent if you can identify the top 10 topics of research in the papers you reviewed (e.g. disease type, interventions, or any unique problems solved by these studies).
- The limitations section needs to be elaborated as there are many weaknesses inherent to review paper designs.
- A key missing component is : why was the study done? what are the implications? how will your study impact practice, research, or policy? What recommendations do you have to change the situation?
Author Response
Please see the attachment which has a cover letter to the Editor and the point by point responses to both reviewer 1 and reviewer 2

Reviewer 2 Report
Abstract. Authors should describe what SORT IT is and provide more background.
The same in the introduction section.
Study design: is it a cross-sectional study or a systematic review
Author Response
Please see the attachment which has a cover letter to the editor and a point by point response to reviewer 1 and reviewer 2

Round 2
Reviewer 1 Report
Thank you for the revisions.
Author Response
Thank you for accepting our revisions.